

# Relevance of warm air intrusions for Arctic satellite sea ice climatologies

Rostosky Philip[1] and Spreen Gunnar[1]

[1]University of Bremen - Institute of Environmental Physics

**Correspondence:** Rostosky, Philip prostosky@iup.physik.uni-bremen.de

**Abstract.** Winter warm air intrusions entering the Arctic ocean can strongly modify the microwave emission of the snow/ice system due to temperature induced snow metamorphism and ice crust formations e.g., after melt-refreeze events. Common microwave radiometer satellite sea ice concentration retrievals are based on empirical models using the snow/ice emissivity and thus can be influenced by strong warm air intrusions. Here, we carry out a long-term study analyzing 41 years of sea

ice concentration observations from different algorithms to investigate the impact of warming events on the retrieved ice concentration. Our results show that three out of four analyzed sea ice concentration retrievals underestimate the sea ice concentration during warm air intrusions which increase the 2 m air temperature above $-5°C$.

This can lead to sea ice area underestimations in the order of $10^4$ to $10^5 \, \mathrm{km}^2$. If the 2 m temperature during the warm air intrusions cross $-2°C$, all retrieval methods are impacted. Our analysis shows that the strength of these strong warm air

intrusions increased in recent years, especially in April. Within the scope of future climate change, it is expected that such warm air intrusions will occur even more frequent and also earlier in the season and thus the influence of these warm air intrusions on sea ice climatologies will become more important in future.

## 1  Introduction

Sea ice concentration estimates from passive microwave satellite observations are an important data set for observing and un-

derstanding the Arctic climate system and for monitoring its recent drastic changes (e.g., Perovich et al., 2017; Maslanik et al., 2011; Kumar et al., 2010). Long term, inter-calibrated time series of sea ice concentration, so called climate data records (CDR) provided by the National Snow and Ice Data Center (NSIDC) (Meier et al., 2021) and the Ocean and Sea Ice Satellite Application Facility (OSI-SAF) (Lavergne et al., 2016, 2019), are of special value since they provide consistent, more than 40 years long time series of Arctic sea ice cover. They naturally have a rather coarse spatial resolution in the order of $25 \, \mathrm{km}^2$. Methods

like the ASI algorithm (Spreen et al., 2008) utilizing observations at a higher microwave frequency at 89 GHz (available since 2002 for AMSR-E/2) take advantage of their higher spatial resolution ($6.25 \, \mathrm{km}^2$ grid resolution for AMSR-E/2).

Even after decades of research, accurate estimation of sea ice concentration and thus sea ice area remain challenging, especially in summer (Ivanova et al., 2015; Liu and Curry, 2003; Tonboe et al., 2003, e.g.,). In winter and spring, atmospheric warm air intrusions, which increased in intensity and frequency in recent years (Graham et al., 2017), can influence not only

the sea ice conditions due to dynamic and thermodynamic processes (Aue et al., 2022; Clancy et al., 2021; Graham et al.,



2019b) but also influence the microwave signal of the snow/ice system due to rapid snow metamorphism and melt-refreeze events (Comiso et al., 1997; Drinkwater et al., 1995; Rückert et al., 2023; Tonboe et al., 2003). Consequently, warming events can impact the quality of satellite sea ice concentration retrievals. Rückert et al. (2023) investigated the impact of such a warm air intrusion and found a strong drop in retrieved ice concentration due to the formation of large-scale surface ice glazing of

the snow.

The aim of this study is to I) analyze how strong the impact of winter warm air intrusions on the retrieved sea ice concentration is and II) whether the impact is increasing in recent years due to an increase frequency of strong warm air intrusions entering the Arctic Ocean. We hereby analyze four common sea ice products: The NSIDC (Meier et al., 2021) and OSI-SAF (Lavergne et al., 2016, 2019) climate data records (CDR), as well as the ASI (Spreen et al., 2008) and the NASA-Team (here

used: the product provided within the NSIDC CDR) sea ice concentration algorithms. The first two are climate data records that means they provide a consistent, quality proved long-term time series of sea ice concentration. The ASI algorithm utilizes higher frequencies and thus a higher spatial resolution and is available since 2002.

The article is organized the following. In chapter two, the physical background of microwave emission of the snow-sea ice system is briefly discussed. The four satellite-based sea ice algorithms and other auxiliary data are introduced. The method to

detect and define warm air intrusions is described in chapter three. In chapter four, the main results are presented and in chapter five, the uncertainty and possible error sources of the results are discussed. The article closes with a conclusion.

## 2    Passive microwave sea ice concentration algorithms

Passive microwave satellite sensors are a common tool to observe the sea ice in the Arctic and Antarctic ocean. To first order, sea ice concentration algorithms take advantage of the strong emissivity difference between sea ice ($\epsilon > 0.8$, depending on

the frequency) and sea water ($\epsilon < 0.3$), which make the two quantities easily distinguishable. In addition, the polarization difference and emissivity changes with frequency can be used to distinguish ice and water. Different algorithms use different methods and frequencies to relate the observed emissivity to sea ice concentration.

In general, the emissivity of sea ice depends on the physical quantities of the ice and snow as well as on the microwave frequency. In Spreen et al. (2008), Figure 1, the typical emissivity of different surface types (first-year ice, multiyear ice and

open ocean) are shown in dependence of typical microwave frequencies used by satellites and most of the common sea ice concentration retrievals.

However, several studies have shown that strong weather events like warm air intrusions, introducing snow metamorphism, melt-refreeze events or liquid water formation in the snow, influence the emissivity of the snow/ice system (Liu and Curry, 2003; Rückert et al., 2023; Stroeve et al., 2022; Tonboe et al., 2003, e.g.,). The emissivity of the snow/ce system depends on

many parameters. However, the main drivers are snow/ice temperature, ice type and the snow microstructure. Ice layers within or ice crusts at top of the snowpack can influence sea ice concentration retrievals that use polarization differences or ratios (due to their strong impact on horizontal polarization, Comiso et al. (1997); Mätzler et al. (1984)). At frequencies higher than 19 GHz, also parameters like snow grain size and shape become important influencing, e.g, retrievals that use gradient ratios



of two different frequencies. Strong changes in the above mentioned parameters influence these quantities and thus introduce
false changes in the retrieved SIC (Tonboe et al., 2003, e.g.,). Figure 4 in (Rückert et al., 2023) (see also Figure A7) shows the
temporal evolution of brightness temperatures and derived parameters for a warm air intrusion in April 2020. A strong impact
on the brightness temperatures at higher frequencies (19 GHz – 89 GHz) is visible during the events while the polarization
difference and ratio (which are used in the sea ice retrievals analyzed in this study) show a strong increase just after the warming
events due the formation of a large-scale glazed ice layer on top of the snow. The gradient ratio shows a strong increase during
the events. It remains higher after the event indicating that snow metamorphism led to stronger scatterer influencing the higher
frequencies. In this study, as already mentioned, four different sea ice concentration products are analyzed. The Ocean and
sea ice facility (OSI-SAF) (Lavergne et al., 2016, 2019) and National Snow and Ice Data Center (NSIDC) (Meier et al., 2021)
algorithms are so called climate data records (CDR) and provide a stable time series of more than 40 years. In addition, the
NASA-Team algorithm (Cavalieri et al., 1997) is analyzed since it is frequently used and is part of the NSIDC CDR. All these
retrievals utilize microwave frequencies between 6.9 GHz and 36.5 GHz. Additionally, we analyze the performance of the ASI
algorithm (Spreen et al., 2008) which is based on 89 GHz. All algorithms are re-sampled to a $25 \cdot 25\,\mathrm{km}^2$ polar stereographic
grid which is the resolution of the coarsest product.

### 2.0.1  NSIDC climate data record

More than 40 years of consistent sea ice concentration observations from various satellites is provided by the NSIDC (Meier
et al., 2021). Daily sea ice concentration is estimated from combining two different products, i.e. the NASA-Team (Cavalieri
et al., 1997) and Bootstrap (Comiso, 1986) algorithms. For the final product, the higher sea ice concentration of the two sub-
algorithm is used on a grid-cell level. The algorithm is based on brightness temperature observations at 19 GHz and 37 GHz
at both polarizations horizontal (H) and vertical (V). A full description of the product can be found at https://nsidc.org/data/
g02202/versions/4. The resolution of this product is $25 \cdot 25\,\mathrm{km}^2$ An uncertainty estimation is given for an individual grid cell
based on the spatial variability within the 9 surrounding grid cells of both sub-algorithms.

### 2.0.2  OSI-SAF climate data record

OSI-SAF provides daily sea ice concentration based on a dynamic algorithm (Lavergne et al., 2016, 2019). Brightness tem-
peratures at 19V, 37V and 37H are used to estimate daily sea ice concentration values at $25 \cdot 25\,\mathrm{km}^2$ resolution. The un-
certainty in sea ice concentration is estimated combining the uncertainty of the algorithm itself and the "smearing" un-
certainty due to the daily ridding of several orbits for the final product. A full description of the product is provided at
https://osi-saf.eumetsat.int/products/osi-450.

### 2.0.3  ASI

The ASI algorithm estimates the sea ice concentration from passive microwave satellite observations at 89 GHz. It is based
on a empirical equation relating the polarization difference (i.e., brightness temperatures at V-Pol − H-Pol) to sea ice con-



centration (Spreen et al., 2008). The spatial resolution of this product is 6.25·6.25 km$^2$ Additional weather filters using lower
frequencies are applied to reduce the impact of clouds and water vapor over the open ocean on the retrieved ice concentration.
The uncertainty is dependent on the ice concentration and is around 5% in the high ice concentration regime.

## 2.1 NASA-Team

In addition, we analyze the NASA-Team ice concentration algorithm (Cavalieri et al., 1997) which is provided within the
NSIDC CDR data set. It is based on a combination of polarization ratio at 19 GHz and the gradient ratio of 37 GHz and 19 GHz
(V polarization). No additional uncertainty estimation for the NASA-Team sub-algorithm is provided.

### 2.1.1 Auxiliary Data

In this study, 2 m air temperature from ERA-5 (Hersbach et al., 2020) reanalysis data are used to detect warm air intrusions.
Here the daily maximum of the temperature is used. ERA-5 is known to have a positive temperature bias in the Arctic (Her-
rmannsdörfer et al., 2023, e.g.,). Possible consequences are discussed in section 5.1.

## 3 Warm air intrusion detection algorithm

A detailed description of the algorithm is given in Appendix A, here only a brief overview of the most important details is
provided. The algorithm is build to detect connected areas where the 2 m air temperature (from ERA5; Hersbach et al. (2020))
crosses a certain threshold within a given time window. Table 1 summarizes the conditions that need to be fulfilled for the
warm air intrusions being considered in this study. An initial temperature threshold is set at $-10°$C. In addition, the duration
of the wave during which the temperature threshold must be at least two days in order avoid short-term fluctuation around the
threshold close in the marginal ice zone. A minimum $\Delta$T of 5 K between the peak temperature and background temperature
is chosen to ensure that a real warm air intrusion is detected and exclude cases were the temperature only fluctuates around a
threshold (e.g., in late spring where the average temperatures can easily be above $-10°$C). Similarly, a minimum size of the
detected area of 200·200 km$^2$ is required. We use April 30 as the end of the algorithm since afterwards, average temperatures
are too high for reliable warm air intrusions detection. Last, a window of 30 days is chosen, which ensures that the whole warm
air intrusion is captured.

The procedure of the algorithm is summarized in Figure A1. I) We apply the algorithm for a period of 30 days and then
compute the whole time series from November 1 to April 30 for each winter season with a slicing window of 5 days (i.e.,
the detection algorithm is run for the time period November 01 to November 30, then for the time period for November 06 to
December 05, etc.). II) For each 30-day period, areas are detected where the temperature crosses at least one of the following
thresholds: $> -10°$C, $> -5°$C or $-2°$C.

In the following, these thresholds are referred to as warming events of category 1 to 3. We do not expect strong snow
metamorphism for category 1 waves and thus changes in sea ice concentration within this area are likely due to sea ice



**Table 1.** Parameters for the initial warm air intrusions detection. $T_{max}$ is the threshold of the temperature maximum that needs to be crossed to trigger the algorithm. $T_{th}$ is the threshold temperature of the three different warm air intrusion categories. $\Delta T$ is the temperature difference between the $T_{max}$ and the background temperature (i.e., the average temperature prior and after the warming event). A minimum size of $200 \cdot 200\,\text{km}^2$ (of connected warming) is required. Time period describes the period where the algorithm is applied and window the range of days the data is analyzed for.

| Quantity | Condition |
|---|---|
| $T_{max}$ | $> -10°\text{C}$ |
| $T_{th}$ | $> -10°\text{C}\, /\, > -5°\text{C}\, /\, > -2°\text{C}$ |
| Duration | $> 2$ days |
| $\Delta T$ | $> 5\,\text{K}$ |
| Size | $200\,\text{km}^2$ |
| Time period | Nov 01 - Apr 30 |
| Window | 30 Days |

dynamics. Assuming similar sea ice dynamics across the three different categories, a stronger drop in SIC in category 2 or 3 compared to category 1 is consequently likely due to snow metamorphism. III) Several corrections and additional checks are applied (see appendix A) and the so called "effective area loss" is calculated based on the reduction of sea ice concentration due to the warming event (see appendix A2).

## 4 Results

### 4.1 Example of Warming Events

In this section, two strong warm air intrusions are discussed in detail. These two exceptional warm air intrusions crossed the majority of the sea ice covered Arctic in April 2015 and April 2020 (Figure 1 and 2). For both warm air intrusions, temperatures crossed $-5°\text{C}$ for a large fraction of the area influenced by the warming wave (orange) and $-2°\text{C}$ in some areas (red). Figure 1 top left shows the maximum extent where the warming peak crossed the different temperature categories. White color refers to the sea ice cover which is not influenced by the warm air intrusion and blue to the open ocean. In the first row, panel 2 to 5 show the sea ice concentration from four different algorithms, averaged over the whole area affected (panel 2), the area with $-10°\text{C} < T < -5°\text{C}$ (panel 3), the area with $-5°\text{C} < T < -2°\text{C}$ (panel 4), and the area with $T > -2°\text{C}$ (panel 5), respectively. In the second row, the left panel shows the average air temperature for the whole area influenced by the warm air intrusion (i.e., where the warming crossed $-10°\text{C}$), with gray shading indicating the 10 and 90 percentiles. Row 2 to 5 show sea ice concentration retrieved by the different algorithms. The day shown refers to the minimum sea ice concentration in the $-5°\text{C} < T < -2°\text{C}$ category. This day can be different for the individual algorithms (see individual maps).

At its maximum extent, the warm air intrusion covered the Fram Strait as well as most parts of the central Arctic, overall, an area of $337 \cdot 10^4\,\text{km}^2$. The strongest drop in sea ice concentration by the OSI-SAF and NASA-Team algorithms happened



in the area where the temperature crossed $-2°$C. However, the area covered by this category is small. The overall impact on the sea ice area is largest for category 2 for all algorithms. The effective area loss (i.e., the overall reduction of sea ice area due to sea ice concentration underestimations; see appendix A2) is calculated for the days from April 16 to April 26 and is $52 \cdot 10^4 \, \text{km}^2$ for the NASA-Team, $75 \cdot 10^4 \, \text{km}^2$ for the OSI-SAF and $112 \cdot 10^4 \, \text{km}^2$ for the ASI algorithm. The overall loss for the NSIDC CDR is small ($< 8 \cdot 10^4 \, \text{km}^2$). For individual days, the effective area loss can be up to $9 \cdot 10^4 \, \text{km}^2$ for the OSI-SAF

algorithm and $19 \cdot 10^4 \, \text{km}^2$ for the ASI algorithm. For the first category, i.e. at $-10°$C, the influence of the warm air intrusion is small for all retrieval algorithms (e.g., $< 10 \cdot 10^4 \, \text{km}^2$ for the OSI-SAF algorithm).

     Figure 2 shows the same as Figure 1 but for a warm air intrusion reaching the Arctic in April 2020. This warm air intrusion is described in detail in Rückert et al. (2023). The authors attributed the false reduction in sea ice concentration to the formation of a large-scale glaze ice layer, lasting more than a week after the warming event. The real sea ice concentration remained

close to 100% in the Central Arctic during and after the warming events. In the warming category 2, the performance of the different algorithms is similar to the 2015 warm air intrusion discussed before, except for the NASA-Team algorithm, which shows a stronger response compared to the previous discussed warm air intrusion. It is interesting to note that while the sea ice concentration starts recovering for the ASI algorithm towards the end of April, both the NASA-Team and OSI-SAF SIC remains reduced indicating that the effect of the warming event might last longer for these algorithms. In the third category,

the drop in sea ice concentration is most pronounced for the OSI-SAF and NASA-Team algorithms. Similar to the example from 2015, the overall loss of the NSIDC product is much smaller ($40 \cdot 10^4 \, \text{km}^2$) than for the other algorithms: $112 \cdot 10^4 \, \text{km}^2$ for OSI-SAF, $120 \cdot 10^4 \, \text{km}^2$ for ASI and $132 \cdot 10^4 \, \text{km}^2$ for the NASA-Team algorithm.

## 4.2   Statistical results

In this section, we analyze the impact of all warm air intrusions detected during the 41-year period from December 1979 to

April 2020. The results are shown in terms of effective area reduction caused by the influence of the warming events. A full definition of this parameter is given in appendix A2. Here we want to remind the reader that this effective area reduction is mainly due to too low retrieved sea ice concentration during (and after) the warming events. As described in appendix A, best efforts have been made to exclude other effects (ice breakup, polynya opening, melting) that could lead to a natural decrease of sea ice concentration. The influence of potential natural processes that reduce the ice concentration during a warm air intrusions

are discussed in section 5.3. A full overview of all warm air intrusions detected during the 41 year period is given in Figure 3, left. Figure 3, right shows only the largest warming events (65 percentile of largest effective area loss) of the OSI-SAF algorithm. The individual dots represents the effective area loss due to the individual warm air intrusions for the three sea ice concentration products products (the NASA-Team algorithm is not shown since it performs similar to the OSI-SAF algorithm). The black line shows the number (#) of warm air intrusions detected for one season. While there is only a slight increase in the

number of overall events (left), there is an increasing trend (0.43 events/decade) of large events (right), which is, however, not significant (p value $> 0.05$) due to the large inter-annual variability.

     Table 2 summarizes the average area affected by warming events for the three different categories as well as the average effective area loss of the different sea ice concentration retrievals. In total, 723 warm air intrusions are detected. The number



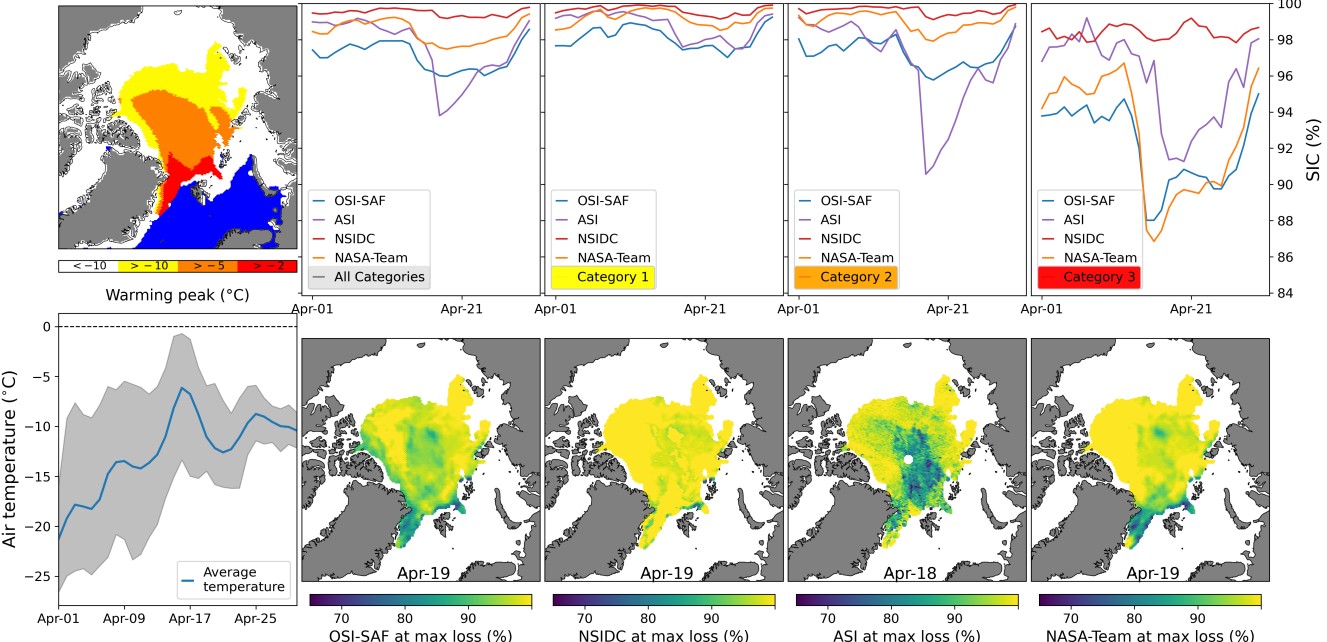

**Figure 1.** April 2015 warm air intrusion and its influence on the four different sea ice concentration products. Top panel shows the extent of the warm air intrusion subdivided into three temperature categories, i.e. $-10\circ$C, $-5\circ$C, and $-2^{\circ}$C (left) for the time period between April 01 and May 01. and the average sea ice concentration for each category. Bottom panel (left) shows the Average air temperature for the whole area influenced by the warming event (i.e., 2 m air temperature $> -10^{\circ}$C). Gray shading indicates the 10 and 90 percentiles. Columns $2-5$ show the sea ice concentration for the different retrievals for the day when their respective ice concentration for the influenced area is lowest (the minimum ice concentration is calculated for the T$> -5^{\circ}$C category). The date is given in the figure.

of warm air intrusions of category 1 and 2 are similar, roughly 100 warm air intrusions less were detected in category 3.

Overall, the NSIDC CDR product performs best with very little effective area loss. The other three retrievals show quite similar performance. In category 1 and 2, the ASI algorithm shows a slightly stronger impact while for category 3, the NASA-Team and OSI-SAF algorithms have the strongest response. In the warming detection algorithm, we additionally analyze the areas, where the warm air intrusion origin from. Over 70% of all detected waves origin from the Atlantic sector of the Arctic. Overall, small events, mainly occurring close to the ice edge in the Atlantic sector, are most frequent (not shown here).



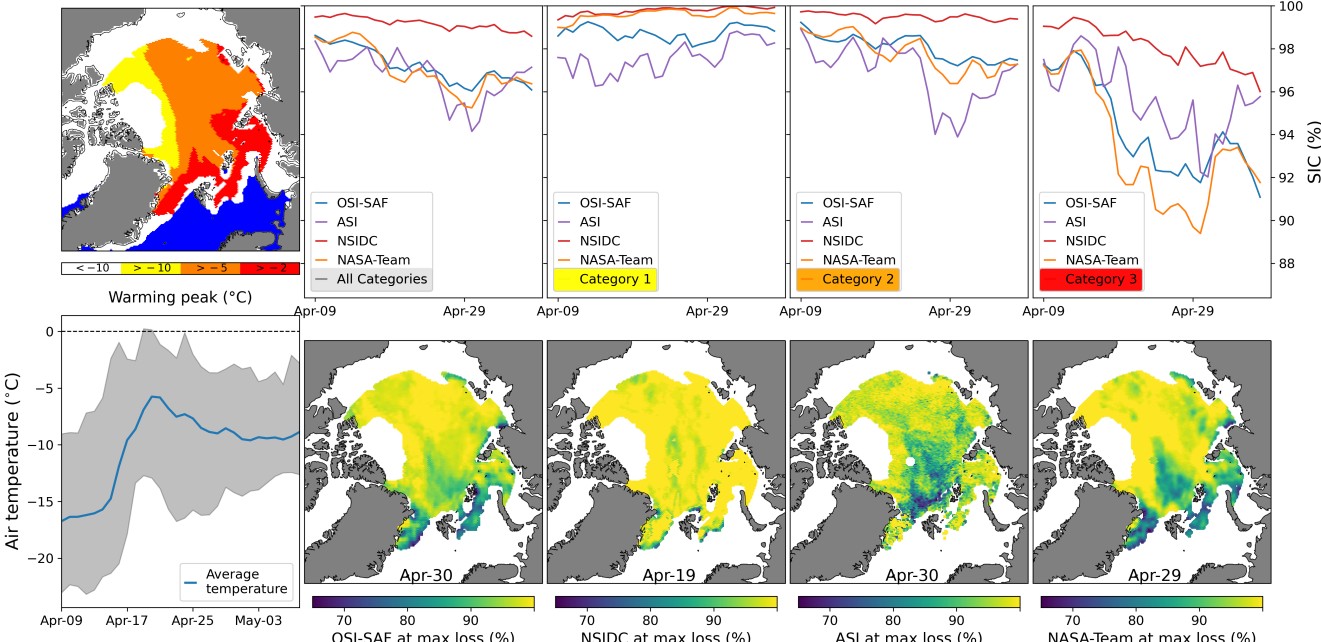

**Figure 2.** April 2020 warming wave and its influence on the three different sea ice concentration algorithms. Top panel shows the extent of the warming wave subdivided into three temperature categories (left) and the average sea ice concentration for each category for the time period between April 09 and May 09. Bottom panel (left) shows the Average air temperature for the whole influenced area (i.e., 2 m air temperature $> -10°$C). Gray shading indicates the 10 and 90 percentiles. Row 2 - 5 show the sea ice concentration for the different retrievals for the day where their respective ice concentration for the influenced area is lowest (the minimum ice concentration is calculated for the T$> -5°$C category). The date is given in the figure.

**Table 2.** Statistical overview of all detected warm air intrusions. Values of averages are given in 1e+04 km$^2$. The values in brackets are the 90th percentile of the data.

| Warming strength | All categories | 1 ($> -10°$C) | 2 ($> -5°$C) | 3 ($> -2°$C) |
|---|---|---|---|---|
| Number of warm air intrusions | 723 | 637 | 640 | 554 |
| Average area (90 pct) | 82 (197) | 40 (88) | 27 (61) | 30 (67) |
| NSIDC area loss (90 pct) | 8 (21) | 3 (8) | 3 (8) | 4 (11) |
| OSI-SAF area loss (90 pct) | 20 (56) | 5 (13) | 6 (19) | 10 (30) |
| ASI area loss (90 pct) | 18 (51) | 7 (17) | 7 (20) | 8 (23) |
| NASA-Team area loss (90 pct) | 21 (56) | 5 (16) | 6 (17) | 10 (26) |





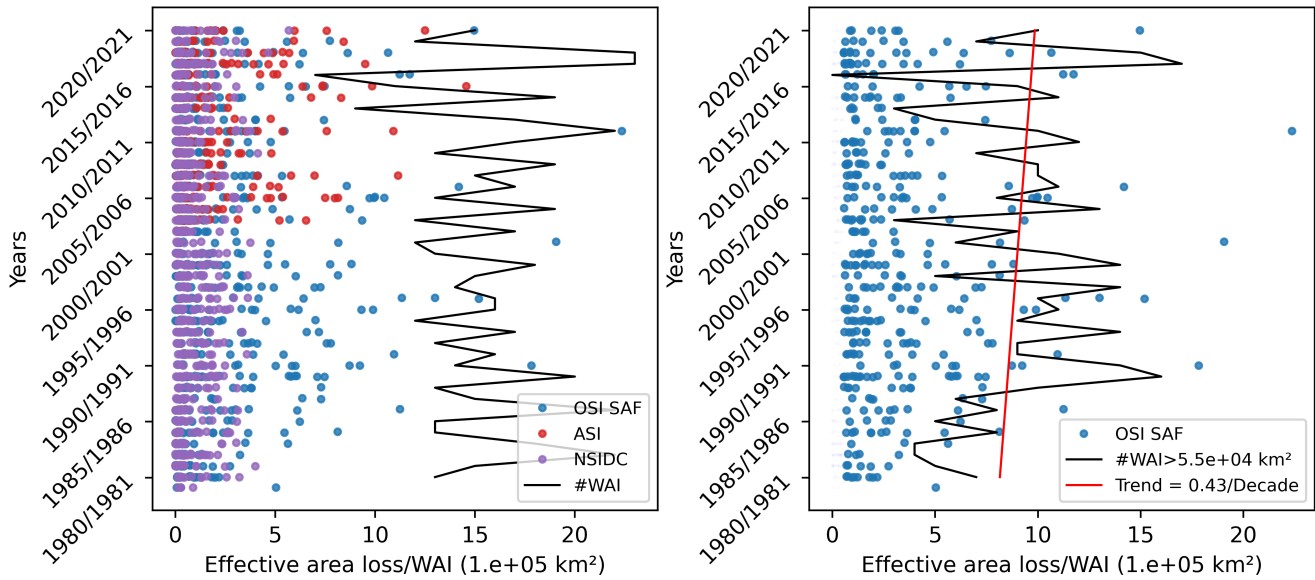

**Figure 3.** Left: Effective area loss of the three sea ice concentration algorithms for each detected warm air intrusion in the 41-years period from November 1979 to April 2020 (the NASA-Team algorithm is not shown since it performs similar to the OSI-SAF algorithm). In addition, the number of warm air intrusions (WAI) per season is shown as black line. Right: Same as left but only for large ($> 5.5e+04\,\text{km}^2$) warm air intrusions and for the OSI-SAF algorithm. In addition, the trend of # of strong warm air intrusions is given.

The performance of the individual sea ice products is compared in Figure 4. Top left shows a boxplot diagram of the effective area reduction divided by the whole sea ice cover impacted by the warming events (%) for the three categories. For all retrievals, the impact of the warm air intrusions increases with the temperature threshold reached by the warm air intrusion. The NSIDC product is only moderately influenced by the warm air intrusions, only for the 3rd category, the average impact is above 1%. In this 3rd category, the OSI-SAF and NASA-Team algorithms show the strongest impact.

Comparing the time periods from 1979 – 1990 (Figure 4, bottom left) and 2010 – 2020 (Figure 4, bottom right), the average loss is similar for the warm air intrusions of category 1. In the categories 2 and 3, an increased effective area reduction is observed in recent years for the OSI-SAF and NASA-Team algorithms, supporting the increased trend shown in Figure 3, right. For example, the mean (median) effective area reduction for the OSI-SAF retrieval for category 3 is 3.7 (2.6) for the early period and 4.1 (3.6) for the late period. A similar increase is found for the NASA-Team algorithm.

Figure 4, top right shows the 41-year average of effective area loss for the OSI-SAF algorithm for the different months from December to April. In April, the warm air intrusions of category 3 have clearly the strongest impact on the performance of the OSI-SAF algorithm, with an average monthly effective area loss of $15.7 \cdot 10^4$. A similar pattern can be found for the other algorithms (except for NSIDC, see Figure A4). Splitting the analysis of the monthly effective area reduction to the periods 1979-1989 and 2010-2020, the effective area reduction in April for category 3 is twice as large in the later decade (Figure A5).

In other months, the differences are less pronounced.





**Figure 4.** Top left: Boxplot of the effective area reduction (%) of the four sea ice concentration algorithms and the three different warming classes. The boxes show the 25th and 75th percentiles. In addition the medians (black line) and 10th and 90th percentiles (whiskers) are given. Top right: Monthly distribution of the effective area reduction for the OSI-SAF algorithm (average over 41 years). Bottom: same as top left but for the winters 1979-1990 (left) and 2010-2020 (right). Note that no ASI data is available before 2002.



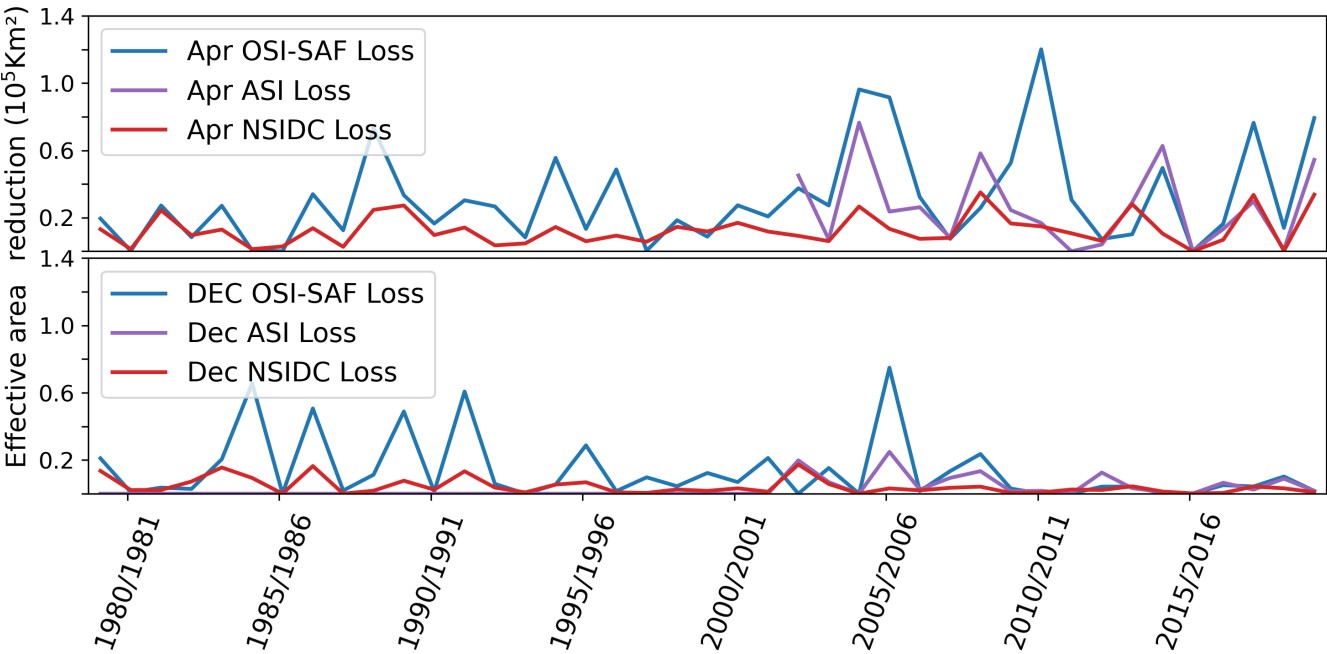

**Figure 5.** Time series of the effective area loss for April (top) and December (bottom) for warm air intrusions of category 3 ($> -2°C$).

Figure 5 shows the time series of the effective area loss during April and December for the OSI-SAF, NSIDC and ASI algorithms for warm air intrusions of category 3. The NASA-Team algorithm is not shown here as it performs very similar to the OSI-SAF algorithm. Note that November, the first month of our analysis, is not shown since during this month, the impact of new forming ice and cracking of ice makes the calculation of the effective area loss more uncertain. In April, most

of the warm air intrusions with strong impact occur in the last two decades, as visible for the OSI-SAF and ASI algorithms. A similar increase of strong warm air intrusions is also found in category 2 (not shown). In December, no increase in effective area reduction, but rather a period with stronger warm air intrusions between 1983 and 1992 is found. After this period, only in a few individual years (e.g., 2007), strong category 3 waves are detected. No increase of warm air intrusions of category 2 or 3 could be found in other months (not shown). We, however, note that in January (February), almost no strong warm air

intrusions of category 3 were detected before 1990 (1986) (not shown).

The results presented in the Figures 3 to 5 show that the strength of the warm air intrusions increased in the last 20 years, especially during April. Especially the impact of warm air intrusions of category 3 strongly increased compared to the earlier years between 1980 and 1990. Not only did the average area of category 3 intrusions increase between the earlier (1980 to 1990 $= 58 \cdot 10^4$ km) and recent years (2010 to 2020 $= 71 \cdot 10^4$ km), but also the average length of these waves increased from

6 to 8 days (not shown).





## 5  Discussion

### 5.1  Uncertainty of the warm air intrusion detection method

An accurate uncertainty estimation of the results provided in the previous chapter is challenging due to the high number of unknowns. A sensitivity test suggest an uncertainty of the method itself of around 5% (see Appendix A3). Additionally, there
are several sources of uncertainty which are hard to quantify. In the following, we will give an overview of the uncertainty sources and provide a qualitative estimate of their impact if possible.

ERA5 2 m air temperature has a known negative bias during cloudy and a positive bias during clear sky conditions over Arctic sea ice (Herrmannsdörfer et al., 2023). Through this bias a possible underestimation of the detected areas where a certain temperature threshold is crossed might therefore lead to an underestimation of the total effective area loss. Also, miss-
classification could be a result of a temperature bias (e.g., due to a negative bias, an area which is classified as $> -10°$C might belong to the $> -5°$C class in reality).

Especially during November, sea ice can rapidly grow also during a warm air intrusion. Consequently, defining the reference period for the effective area loss calculation (see Appendix A2) can be challenging and is sometimes not possible. Therefore, the results presented here are limited to December - April, event though, warm air intrusions frequently occur also during
November. In April, after a warm air intrusion, the air temperature can remain high until the end of the winter season (the detection stops at April 30 but additional 10 days in May are analyzed to properly define warming events occurring in the last third of April). Consequently, it is sometimes difficult to define the end of a warm air intrusion and to detect a proper reference period. These initially detected warm air intrusions were discarded in the results presented here and thus the effective area loss in April might be underestimated.

Strong synoptic events like warm air intrusions can push and/or move the sea ice edge and, due to increased ice dynamics, can lead to the opening/and closing of leads. Leads modify the sea ice concentration derived by passive microwave algorithms and thus can influence the calculated effective area loss. To mitigate this influence, the algorithm is only applied in areas where the sea ice concentration remains above 65% for the whole period of investigation. This is a conservative approach to ensure that there is no false detection of sea ice reduction due to the warm air intrusion but consequently the effective area reduction
considering the whole sea ice covered Arctic can be expected to be higher than presented here.

### 5.2  Uncertainty Estimates of Satellite Sea Ice Concentration Retrievals

Both CDR algorithm provide a sophisticated uncertainty estimation. For the OSI-SAF algorithm, the uncertainty in the central Arctic (i.e., high sea ice concentration) is usually around 2% - 3%. In areas with high ice dynamics (= high temporal variability in ice concentration), the uncertainty can increase to 5%. A similar increase in uncertainty is also visible under the effect of
strong warm air intrusions (e.g., category $> -5°$C or higher). During strong warm air intrusions, sea ice concentration can drop by more than 10% for a substantial amount of grid cells, meaning that this drop in SIC is not fully covered by an increase of uncertainty.



For example, for the two warming events discussed in detail in the results section (Figures 1 and 2) in 50% (Figure 1) and 30% (Figure 2) of the area affected ($> -5°$C, reference SIC $> 95\%$), the reduction of sea ice concentration was larger than the

uncertainty estimate of the OSI-SAF algorithm. Similar results were found for warm air intrusions $> -2°$C in the high sea ice concentration domain. It is to note that large parts of the $> -2°$C warm air intrusions happen in the marginal ice zone in the intermediate sea ice cover domain (50% < SIC < 75%), here the uncertainty of the OSI-SAF algorithm is higher and often as high as the reduction in SIC due to the warm air intrusions.

The NSIDC CDR provides a quality assessment and a standard deviation to estimate the uncertainty of the sea ice con-

centration. The standard deviation of a grid cell is estimated from the 9 surrounding grid cells from each sub-algorithm used (NASA-Team and Bootstrap). In the central Arctic it is usually between 0% and 2% but can be 10% and higher in the marginal ice zone. As shown in Figure 4, the NSIDC CDR is overall less influenced by warm air intrusions and mainly the $> -2°$C warm air intrusions have an impact on the derived ice concentration. For warm air intrusions $> -2°$C in the high sea ice concentration domain, we find that in 25% – 40% of the cases, the sea ice reduction is higher than the uncertainty of the retrieval.

It is to note that the sea ice concentration reduction of individual grid cells is rarely above 5% in the NSIDC CDR.

Overall, the effect of the warm air intrusions is not fully captured by the uncertainty of both CDR algorithms. An increased uncertainty needs to be considered in the case of strong warming events. While the NSIDC CDR generally performs best during warming events, we note that an overestimation of sea ice concentration can be a result of the method applied in this algorithm. Especially in areas like the Greenland Sea, frequent polynyas and large leads open after strong storm events. These

are not captured by the NSIDC CDR, while in the OSI-SAF CDR or the ASI algorithm using its natural resolution ($6.25\,\mathrm{km}^2$), these events are clearly visible (see Figure A6).

The results of this study show that in recent years, high impact warm air intrusions have become more frequent and their impact on satellite sea ice algorithms became stronger (Figure 5). With further Arctic warming it is likely that strong warm air intrusions will occur even more frequent and might extent to earlier months in the year, although no such trend was observed,

for e.g., March in this study. The associated increased uncertainty in winter time satellite derived sea ice concentration needs to be addressed. In principle, a warm air intrusion detection algorithm as presented here could be used to reprocess sea ice climatologies, "correcting" for the impact of strong warm air intrusions. However, currently the high uncertainty of this method due to, e.g., not considering leads, flagging out the marginal ice zone area, choice of the reference sea ice concentration period, make it difficult to reliably "correct" sea ice concentrations. Further improvements of this approach and additional data sets

(e.g., SAR lead classification) are needed. A more simple approach could aim for including air temperature variability in the uncertainty estimation. This could help to improve the winter time uncertainty estimation, especially in the Central Arctic where strong warm air intrusions are one of the main sources for uncertainty.

## 5.3 Sea Ice Concentration Reduction

Aue et al. (2022) analyzed the impact of cyclones on the sea ice concentration in the Atlantic sector of the Arctic ocean and

found an average drop in sea ice concentration of less than 2% in ±3 days around the cyclone events. The drop in SIC was





stronger close to the ice edge and in the low ice concentration regimes. Almost no drop in sea ice concentration was found in the central Arctic.

Table 3 shows the cases detected of warm air intrusions (%) during which the average sea ice concentration reduction is above 2% for the three different categories. For warm air intrusions of category 2 or 3, the majority of warm air intrusions 280 (>60%) led to a reduction of SIC >2% for the OSI-SAF and NASA-Team algorithm. In comparison, for the NSIDC and ASI algorithm only about 30% of the warm air intrusions in category 2 caused a reduction of >2%. For category 3, almost 50% of all warming events caused a sea ice concentration reduction >2% for the ASI algorithm which, however, is still well below what was found for OSI-SAF and NASA-Team (roughly 70%).

It is important to keep in mind that in this study, SIC reduction was averaged over the whole affected period which is much 285 longer (on average 12 days) than the 6 days considered in the study by Aue et al. (2022). More than 5 days after the cyclone event, Aue et al. (2022) reported a SIC increase in most of the areas in the Arctic and only an average reduction of <1% in the Greenland Sea. Using a threshold of 1% SIC reduction in our analysis, for OSI-SAF, NASA-Team and ASI over than 80% of the warm air intrusions (for category 2 and 3) caused an average sea ice concentration reduction above the 1% threshold. Only the NSIDC algorithm remains below this threshold in roughly 50% of the cases. An other recent study found similar 290 effects of cyclones on the sea ice concentration (Clancy et al., 2021) leading to a reduction in ice concentration between 1 - 3%, with strongest effects in the marginal ice zone. Graham et al. (2017) found a strong storm introduced reduction of sea ice concentration in February 2015 north of Svalbard during the N-ICE2015 drift campaign. A detailed analysis showed that for the February 2015 warm air intrusion (as detected with the here presented algorithm), the area around (and south of) the campaign location was masked out since the automatic ice edge and polynya masks triggered (see Appendix A). The remaining effected 295 area was of medium size ($5 \cdot 10^5$ km$^2$) and the sea ice reduction (e.g., 2.5% for the OSI-SAF algorithm) was not exceptional. The Aue et al. (2022), Clancy et al. (2021), and Graham et al. (2019a) studies attribute the changes in SIC to natural causes implied by the passing cyclone and not a deficiency of the satellite SIC products used.

These findings demonstrate that in the majority of the warm air intrusions or cyclones, one can expect a natural and real change in SIC of $1 - 3\%$, e.g., by opening of leads. The reduction in sea ice concentration we find for warming events (for 300 some algorithms $> 2\%$ SIC reduction in $> 60\%$ of the cases, see Table 3) is thus more than what would be expected due to potential increased lead openings. This is especially true for the strong warm air intrusions, were we often found the impact of the warming event on SIC lasting up to a week after the event.

In summary, we find that during and after warm air intrusions, the reduction of sea ice concentration by all algorithms (except for the NSIDC CDR) is, in many cases, higher than what would be expected by the opening of leads due to sea ice dynamics.

## 305   6   Conclusions

In this study, we analyzed the impact of warm air intrusions on sea ice concentration climatologies derived from microwave radiometer satellite measurements. We investigated the following questions: I) is the impact of warm air intrusions on satellite-based sea ice concentration algorithms a relevant source of uncertainty for sea ice climatologies? II) In the course of Arctic



**Table 3.** Cases of warm air intrusions during which the average sea ice concentrationdropped by more than 2% for the three different temperature thresholds.

|  | OSI-SAF | NSIDC | NASA-Team | ASI |
|---|---|---|---|---|
| $> -10°C$ | 40% | 11% | 38% | 23% |
| $> -5°C$ | 60% | 32% | 61% | 30% |
| $> -2°C$ | 70% | 34% | 68% | 50% |

amplification, do these warm air intrusions have increased in recent years and will they become more relevant in future?

310

 I) Based on the results presented here (e.g., Figure 1, 2 and 4), we have shown that warm air intrusions can have a strong impact on most sea ice concentration algorithms (except the NSIDC CDR), leading to an underestimation of sea ice concentration and consequently an underestimation of sea ice area. Depending on the algorithm, this underestimation is on average between 2% to 4% of the total area affected by the warm air intrusion (Figure 4) for warm air intrusions of category 3. For large warm

315 air intrusions (e.g., in April 2015, Figure 1), the effective sea ice area reduction is in the order of $75 \cdot 10^4 \, \text{km}^2$ to $115 \cdot 10^4 \, \text{km}^2$ (OSI-SAF and ASI) over the whole influenced period (9 days) and up to $19 \cdot 10^4 \, \text{km}^2$ for a single day (ASI). During strong warm air intrusions, reduction in sea ice concentration due to the warming is often not fully captured in the uncertainty of both CDR products.

 II) In general, we found that warm air intrusions of category 3 (i.e., the warmest $> 2°C$ category) are most prominent in

320 April (Figure 4, top right) and that in April they have an increased impact on the satellite algorithms in the last two decades (Figure 5). For the other months, no such increase is evident in the data. In the scope of a further climate change, it can be expected that such warm air intrusions will become even more frequent and that category 3 intrusions will extend further into the central Arctic.

 During strong warming events the NSIDC CDR benefits from including the Bootstrap algorithm (Comiso, 1986) which

325 seems to be less sensitive to the impact of the warm air intrusions on the snow/ice system. In general, lower frequencies (e.g., 6.9 GHz) are less influenced by warm air intrusions (Rückert et al., 2023), since these are mainly short-term events and the warming rarely reaches the snow/ice interface where the low-frequency signal mainly origins. With future higher-resolution low frequency observations (e.g., the CIMR – Copernicus Imaging Microwave Radiometer) such algorithms using lower frequencies could be used as a reference for correcting current algorithms during strong warm air intrusions.

330 *Data availability.* The NSIDC CDR and NASA-Team sea ice concentrations used in this study are available via https://nsidc.org/data/ g02202/versions/4. The OSI-SAF CDR is available via https://osi-saf.eumetsat.int/ and the ASI ice concentration can be downloaded from https://data.seaice.uni-bremen.de/amsr2/asi_daygrid_swath/n6250/. The ERA-5 2 m air temperature data can be obtained from https://cds. climate.copernicus.eu/cdsapp#!/home.





## Appendix A:  Warm air intrusion detection algorithm

### A1  Overview of procedure

An overview of the procedure of the algorithm to detect warm air intrusions is given in Figure A1. The input for the detection algorithm are 30-days arrays of gridded ERA-5 2 m air temperature (T2m) and OSI-SAF sea ice concentration (SIC). Based on the thresholds defined (here used: category 1: $> -10°$C, category 2: $> -5°$C, and category 3: $> -2°$C), the *Threshold detection* procedure detects connected areas where the temperature thresholds are crossed for a certain amount of days (here

used: 2 days). Several not connected regions can be detected and they will be merged if their borders are separated by 2 pixels ($= 50$ km) or less.

During the detection, a land mask, a polynya mask, and a dynamic ice edge mask are applied. The ice edge mask is derived from the minimum extent of SIC $> 65\%$ during the 30-day period. For the polynya mask, we analyzed 41 years of winter OSI-SAF sea ice concentration and masked out areas where the SIC frequently dropped below $50\%$ while the ice edge was far

away. Masked out areas are, e.g., Svalbard or Novaya Semlya (see Figure A2).

the *Threshold detection* interacts with the the *Similarity check* procedure. Here, it is tested if the current detected warm air intrusions are similar to the previously detected ones. First, the overlapping area of the new and previous warm air intrusion is calculated. If the overlap is above $70\%$, the larger intrusion area is used for further calculations and the other one is discarded. If the extent of both intrusions is similar, the warm air intrusion with the larger effective area loss (see section A2) is used.

In the next step, the detected areas are combined with the sea ice concentration data sets and the procedure *Effective area reduction* is applied. In this module, the effective area reduction (see section A2) is calculated for each sea ice concentration data set and for each defined threshold. The data is finally stored in the *Output array*. In this study, the algorithm is applied for the winter season November – April. With a 5 day time step, in each iteration, 30 days of T2m and SIC are analyzed. A sensitivity test showed that a 5 day step is needed to ensure that the warm air intrusions is optimally captured while the

processing time of a season remains reasonably short.

### A2  Effective area reduction

In order to break down the impact of the warm air intrusions on the sea ice concentration to one number, the "effective area reduction" is defined. This parameter describes the effective area reduction due to a reduction of sea ice concentration during the period impacted by a warm air intrusion in comparison to the reference sea ice concentration before and after the event.

Figure A3 shows the basic principle of this calculation. Shown are the sea ice concentration and the 2 m air temperature averaged over the area where the threshold $> -5°$C was crossed. The whole area was around $2 \cdot 10^6$ km$^2$. The reference sea ice concentration is the average SIC from the days before and after the influence of the warming event on the ice concentration. In this example, the sea ice concentration during the reference period (blue) is above $99\%$ and drops to $95\%$ during the warming event (red). The resulting effective area loss is $6 \cdot 10^5$ km$^2$ over the whole red period (16 days) or $\approx 2\%$/day.

Defining the time period during which the sea ice concentration is affected by the warm air intrusion is not straight forward. In this study, the following procedure was chosen: The start of the affected period is based on the day, when the 2 m air





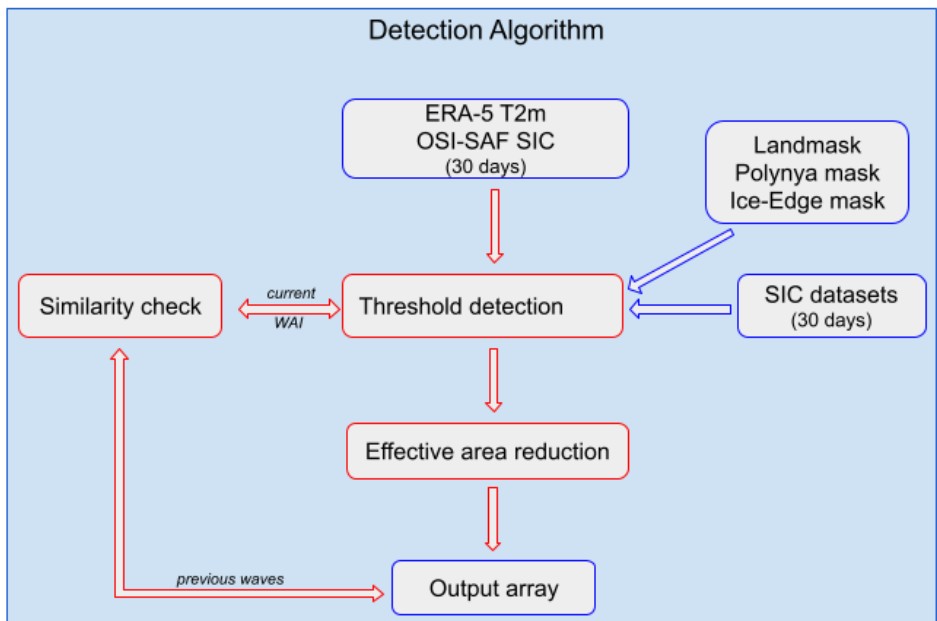

**Figure A1.** Overview of the warm air intrusion detection algorithm. Here WAI is the abbreviation for warm air intrusions.

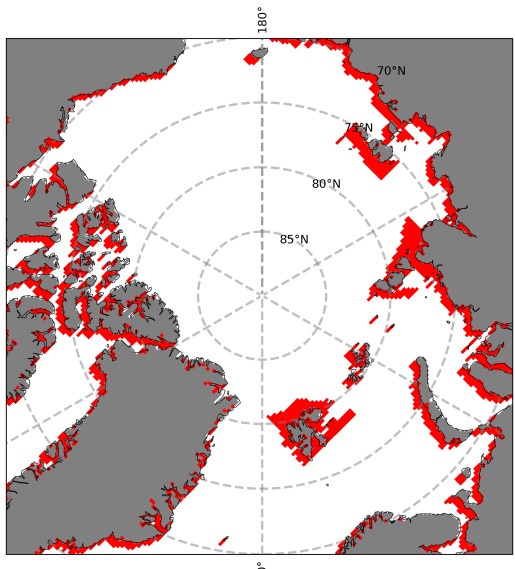

**Figure A2.** Extended landmask used in the algorithm. Grey areas indicate the landmask and red areas indicate the area where the polynya mask is applied.

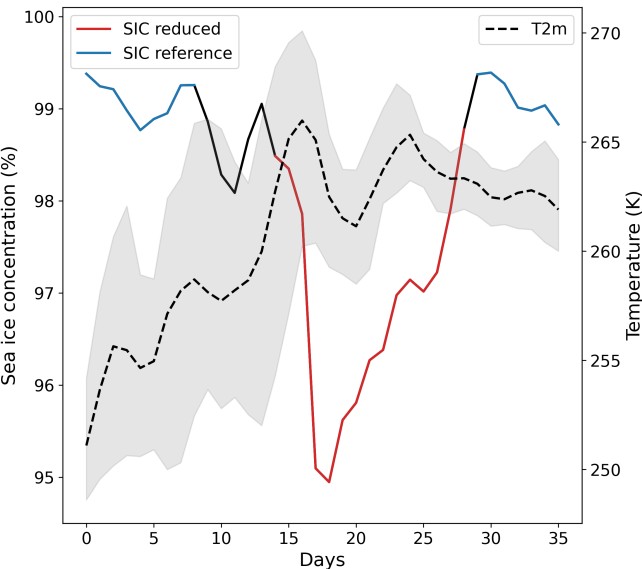

**Figure A3.** Example for the effective area loss calculation. The sea ice concentration is an average over the whole area affected by the warm air intrusion (around $2 \cdot 10^6 \text{km}^2$). Gray shading indicate the 15 and 85 percentiles of the 2 m air temperature.

temperature first crossed the $> -10°C$ mark. The end of the period is reached when the sea ice concentration is close (within 1%) of the reference sea ice concentration from before the event. If the latter does not apply, the end of the affected period is set to 10 days after the peak of the warming.

**A3  Sensitivity check**

We note that the calculated effective area loss is depending on the choice of parameters (reference periods, affected period, iteration step length). We performed a sensitivity study varying the controlling parameters for five test warm air intrusions to estimate their impact on the effective area loss. In the example given in Figure A3, varying the start and end dates for the affected period (red line) by $\pm 2$ days would lead to changes in effective area loss of around 5%. Varying the selection of the

reference period in a similar way led to changes of around 3%. Reducing the iteration step to 1 day instead of 5 days to find the optimal window for the warm air intrusions did not lead to relevant changes for the example shown above. Analyzing 5 selected large warm air intrusions (March 1990, December 2016, April 2015 and 2020, and February 2020), on average, the effective area loss varied by $\pm 6\%$ due to varying the reference periods and affected periods as described above. In two cases, optimizing the warm air intrusion detection by using an iteration step length of 1 day increased the effective area loss by more

than 3% and 7%, respectively. For the other cases, no relevant changes were observed. Concluding, an uncertainty in effective area loss of roughly 5% can be expected introduced by using fixed parameters for the warm air intrusion detection.





**Figure A4.** Monthly distribution of the effective area reduction for the four different algorithms (average over 41 years).



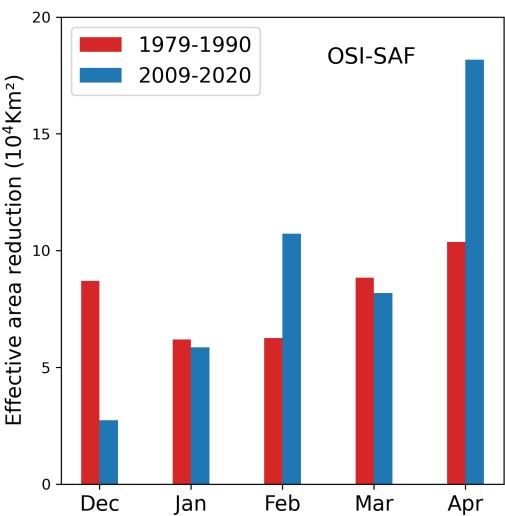

**Figure A5.** Monthly distribution of the effective area reduction for the OSI-SAF algorithm for the years 1979-1990 and 2009-2020 and for warm air intrusions of category 3 (i.e., 2 m temperature $> -2°$C.

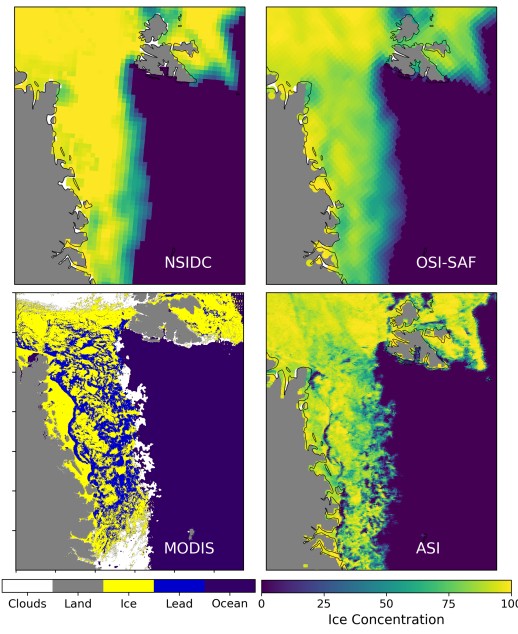

**Figure A6.** Sea ice concentration in the Fram Strait for 20 April 2020 for three different algorithms. In addition, sea ice leads derived from MODIS (Willmes and Heinemann, 2015) are shown.







**Figure A7.** Figure 4 from Rückert et al. (2023): Top row shows the air temperature at 2 m (a). Collocated satellite measurement of TB (b) and PD (c) around *Polarstern* (daily averages). (d) Gradient ratio of 36.5 GHz and 18.7 GHz and polarization ratio of 18.7 GHz.





*Author contributions.* PR performed the study and wrote the initial version of the article. GS contributed to the discussion and development of the study. All authors revised and commented on the article.

*Competing interests.* The authors have declared that no competing interests exist.

*Acknowledgements.* This work was funded by the Bundesministerium für Bildung und Forschung (BMBF) through the IceSense project (grant BMBF 03F0866B). Further support was provided by the European Union's Horizon 2020 research and innovation programme (grant No 101003826) via project CRiceS (Climate Relevant interactions and feedbacks: the key role of sea ice and Snow in the polar and global climate system). We further acknowledge NOAA/NSIDC, Eumetsat OSI-SAF and the Copernicus Climate Change Service for providing the sea ice concentration and air temperature data sets used in this study. Further, we thank Dr. Sascha Willmes for providing daily sea ice lead
data for the Arctic.



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
