# Peer review of "Relevance of warm air intrusions for Arctic satellite sea ice concentration time series"

_The Cryosphere, 2023_

## Referee Comment (RC2)

Review of "Relevance of warm air intrusions for Arctic satellite sea ice climatologies" by Rostosky and Spreen. https://doi.org/10.5194/tc-2023-69

The authors introduce a method to detect and categorize Warm Air Intrusion from atmosphere reanalysis temperature data. They apply the method over four decades and document a (slight) increase in the occurrence and severity of WAIs. They also document that four widely used sea-ice concentration datasets derived from passive microwave data respond differently to WAIs. All datasets are affected by the most severe WAIs (T2m > -2C), but three out of four are already significantly affected by less severe WAIs (-10C < T2m < -5C). The study is of interest for users of sea-ice data records, and can be interesting for developers of sea-ice concentration retrieval algorithms. It can be published after some revisions.

GENERAL COMMENTS:

Kern et al. (2019) concluded that the NSIDC CDR (v3) was high-biased at 100% SIC, while the OSISAF CDR (v2) was low-biased at 100% SIC. This is very well in line with your results. For a SIC CDR to appear un-affected by WAIs, a strategy is to overestimate the SICs (by selecting tiepoints that return > 100% SICs) and apply the 100% SIC threshold. You see this with your Figure A.6. At present, algorithm calibration is mentioned in passing with only 2 sentences (257-261) in your manuscript, although it is possibly a key element of what you observe. By referring to Kern et al. (2019), using your knowledge of how SIC algorithms work and especially the impact of tie-point selection, and moving your Figure A.6 to the main body of the manuscript, I invite you to significantly expand on these aspects in the Discussions.

From your text, it looks like you are using the OSI SAF SIC CDR v2 (OSI-450 and OSI-430-b). If this is the case, I strongly encourage you to update your study to use the SIC CDR v3 (OSI-450-a and OSI-430-a). The v2 CDR is deprecated, the v3 CDR was released in October 2022. Part of the update from v2 to v3 was exactly to reduce the low bias observed in v2 at high SICs. Using the v3 SIC CDR should not be too cumbersome, as it has the same file format and map projection as v2. This would ensure that your manuscript stays relevant for the latest version of the SIC CDRs. If you are alreadt using the v3 CDR, all is good, but you should make it clear in the text (see specific comments below).

If SIC algorithm developers want to improve their SIC CDRs, they will need to study in details how the Tbs change with WAIs on many WAI cases, across satellite missions, etc… Even if imperfect, your WAI detection method could be a key asset to get these studies started. I thus invite you to: 1) publish the maps of your WAI detection (e.g. daily maps with integer values recording if there was a WAI, and what category it was) and 2) publish the software for the WAI detection on a software repository. This will ensure not only transparency and reproducibility of your research, but also help future developments of SIC algorithms to improve on the situations you report.

SPECIFIC COMMENTS:

Title : Your study is only about Sea Ice Concentration (not Drift, Type, etc…). Also, "Climatologies" often refers to the average (or median, etc…) over long time periods (WMO definition). The data you are studying are often referred to as "Climate Data Records", or "timeseries". Please modify the title to reflect the above.

Abstract :
- "during warm air intrusions". This does not convey that the SIC is impacted after as well (when when Temps are back to nominal).
- The last sentence could be made shorter and more impactful.

L23 : You could have cited the recent Kern et al (2019, 2020, 2022) papers. Especially Ker et all 2020 look specifically at summer conditions.

L25-26 : Here would be a good place to remind the readers that WAIs have different phases, and that the effects on the microwave emissions might perdure several days after the WAI is finished (in terms of higher temperature).

L29 : surface ice glazing was the main hypothesis of Rückert et al. (2023) but formulate your sentence as a certainty. You could revise. Also it might be interesting to note that the two case studies in Rückert et al. (2023) were along the MOSAiC drift.

L33-34 : please specify which versions of the CDRs you are using. Also you should spell out these acronyms.

L38 : "the following" → "as follows". Possibly rework these sentences. For example you state here that chapter two will briefly discuss the physics of microwave emissions, but the title of the section is about SIC algorithms, which is not exactly the same thing. Also, I think these are called "Sections" and not "Chapter".

L45 "quantities" → "surfaces"

L53 You could rework the transition from your first to second paragraph. For example open the 2nd with "However, the emissivity of the snow/ice system depends on many parameters". And maybe introduce the WAIs later in the 2nd paragraph, when the other parameters have been discussed.

L73, L81, L85, L93, L97, etc… check the numbering of your headings. Should it be "2.1", "2.2", etc…

L74 From the URL it seems you are using NSIDC SIC CDR v4. Please check and write it in the text.

L81 OSI SAF is prefered to OSI-SAF (throughout the text and figures).

L82 From the URL it seems you are using OSI SAF SIC CDR (OSI-450) and ICDR (OSI-430-b) v2. Check and write it in the text.

L82 The OSI SAF processing chain also uses ERA5 NWP fields and corrects the Tbs for the atmospheric effects. This could be noted as one could have assumed that this would help in the early phases of the WAI.

L87 I thought ASI was the name of an algorithm. Could you write "ASI SICs from AMSR-E and AMSR2", e.g. ?

L97 Since your only auxiliary data is ERA5, you can rename your section.

L98 I think ERA5 (not ERA-5) is the accepted acronyms.

L100 For the ERA5 bias, please add some more citations rather than the etc, e.g. Wang et al. 2019, Batrak and Müller (2019).

L119 the definition of the three categories is not entirely clear. L118 you define T threshold with > -10C, > -5C, and > -2C. Thus, a category 2 (> 5C) is automatically also a category 1 (since >10C). But later (L133) you use the thresholds as brackets ( -10C < T < -5C). Please clarify in the text and review your figure captions and headings of Table 2.

Fig 1 and Fig 2 : Please use the same y-range for the top row graphs. Currently Fig 1 uses (84-100) while Fig. 2 uses (86-100)

Table 2: In Fig 1, Fig 2, and Table 2 you introduce the concept of "All Categories" but this is not explained in the text. Is it the collection of all Cat1, Cat2, and Cat3 events? But it is then not obvious why the number of "All categories" events in Table 2 is not the sum of the three other Categories.

Table 2: specify the units of the area and area loss.

L180: Rather than performance of the algorithm, you could refer to the impact of the WAIs on the algorithms.

Fig 4 : Fix NSIDC (not NSICD) and OSI SAF (not OSI-SAF).

Fig3 and Fig 5 : re-use the same colors as Fig 1 and Fig 2 (for the data sources)

Fig 4: use the same colors as Fig 1 and Fig 2 (Categories)

Fig 4: add text for the time periods covered in the panels.

L206: From what I understand, you have two observations: 1) the number / area / category of WAIs detected by your method increases throughout the 40 years, and 2) their impact on the SIC products increases. You might want to reformulate this paragraph to make this distinction.

L218 : The sentence starting with "Through…" seems broken. Please rework.

L237 : The sentence would work equally well without "sophisticated".

---

## Author Comment (AC1)

**1 Reviewer 1**

Relevance of warm air intrusions for Arctic satellite sea ice climatologies

Rostosky and Spreen

This paper investigates the impact of Arctic warm air intrusions (WAIs) on four passive microwave sea ice concentration products. To complete this analysis, the authors developed a methodology to detect WAI events from ERA-5 reanalysis temperatures and classify their intensity into three categories. Results show that the most extreme warm air intrusions reduce sea ice concentrations in most data products leading to an underestimation of sea ice area within the affected area of 2-4%. Further, the authors demonstrate a non-significant (due to high interannual variability) increase in the frequency of WAI events with a peak occurrence of the most severe events in April.

I think this paper makes a good contribution to the literature and makes the case that these WAI events have a significant impact on passive microwave sea ice concentration retrievals, the effects of which are expected to increase due to climate change. The analysis is thorough and appropriate for publication in The Cryosphere. I do think the paper would be improved with minor revisions to clarify some points of the methodology as I describe in my comments below.

*We thank the reviewer for their positive and helpful comments. We believe that, thanks to these reviews, the manuscript has improved a lot. Please find below our responses to the individual comments. Note that Reviewer 2 pointed out that an updated version of the OSI-SAF CDR (OSI-450-a, version 3) is available. In the revised version of the manuscript, we updated our analysis using the OSI-450-a. While some numbers slightly changed, the outcome of this study remains the same.*

General Comments

"... climate data records (CDR) and provide a stable time series of more than 40 years." (L68): Other SIC data sets that are not CDRs also provide stable, long time series. The big thing that differentiates the NSIDC CDR from, for example, NASA-generated SIC products is that as a CDR, the NSIDC product does not involve any manual corrections that cannot be reproduced exactly by the code. This is the important distinction that defines how a CDR is different from any other long data time series. For this study, I don't think that the distinction between CDRs and other time series of SIC is as important as the text would indicate.

*We thank the reviewer for this valid point. We will reformulate the description of the CDRs and clarified their relative importance for this study. We e.g., will add*
*"CDRs are designed to provide consistent, reproducible long-term timeseries of climate variables. Long term sea ice concentration is also provided by e.g., the NASA-Team algorithm which is frequently used and is part of the NSIDC CDR."*

"... the NSIDC CDR benefits from including the Bootstrap algorithm..." (L324-325): I was waiting for this point to be stated throughout reading the whole paper, but it does not appear until the very last paragraph and is only one sentence. I think a slightly longer explanation for the reason why the NSIDC product is performing better than the other is necessary for the readers especially since the NASA Team algorithm (from within the NSIDC CDR) performs very differently than the NSIDC CDR throughout your analysis. Specifically explain (1) that the NSIDC CDR SICs are primarily sourced from the Bootstrap algorithm during the WAIs and (2) some details on how the Bootstrap algorithm avoids the extreme sensitivity to the WAI events seen in the other algorithms (e.g., daily dynamic tie points, etc.). Point 2 would explain the "why" for point 1. This discussion should be moved to Section 5, and not be introduced in the conclusions.

*We will introduce the Bootstrap algorithm in the data section and will add a paragraph about the performance of the NSIDC CDR in the discussion section*
*"Since the NSIDC CDR is less influenced by warm air intrusions, it is worth discussing the reason for its performance. The*

*NSIDC CDR is computed from the NASA-Team and Bootstrap algorithms. The CDR sea ice concentration is based on the sub algorithm with the higher sea ice concentration, which, in case of strong warm air intrusions, is the bootstrap algorithm (since the NASA-Team shows a strong underestimation of sea ice concentration during warm air intrusions). In the NSIDC CDR, an updated bootstrap algorithm with dynamic (daily adapted) tie points for open ocean and full sea ice cover is used (Comiso et al., 2017). By using dynamic tie points the impact of changing snow and surface conditions are mitigated and thus the impact of warm air intrusions on the derived sea ice concentration is reduced."*

I suggest that the authors consider moving the algorithm details from appendix A1 and A2 to the methodology (section 3). As it is now, neither section 3 nor the appendices are complete descriptions of the method, and some information is repeated in both places.

*We agree with the reviewer and will merge the appendix A1 and A2 with section 3. In addition, we will restructure parts of section 3 to improve its readability*

Specific Comments

L99: Here you state that you use the daily maximum 2m air temperature, however, throughout the rest of the paper you only refer to 2m air temperatures. These are not the same. I suggest changing your wording when mentioning 2m air temperatures to clarify that it is the daily max 2m air temperatures instead.

*We will now always refer to the daily max. 2 m air temperature*

Figure 2 and L147-157: There is  10 days difference between the max loss days between the NSIDC and other three algorithms. Why is this the case when the peak temperature is 19 April? The max loss dates from case 1 (Figure 1) are much closer (only 1 day difference). Can you add some commentary to this paragraph explaining why case 2 has a much longer time difference in max loss days?

*Figure 2 highlights the case that the maximum of sea ice underestimation does not necessarily occur on the the same day as the maximum of the warming. A unique feature of this particular warming event was the formation of large scale surface glazing happening after the warming event (refreezing). Details are described in Rückert et al 2023 (submitted), the study is available here: https://doi.org/10.31223/X5VW85. Note that the NSIDC CDR is almost not affected by this WAI (at category 2) and thus the date of maximum sea ice reduction is not meaningful for this algorithm. Also for Figure 1, the day of maximum loss is not that meaningful for the NSIDC retrieval since the loss is overall very low. We therefore mainly focus on the day of the three other algorithms. We will add*
*"In both examples, the day of minimum sea ice concentration is consistent within the products except for the NSIDC algorithm. However, the overall sea ice concentration reduction for the NSIDC CDR is very low and thus the result are less meaningful for the NSIDC retrieval."*

Comments on Appendix A1: (L347) How many days are between the "new and previous warm air intrusions"? If I understand the procedure correctly (L346-349), there can only be one WAI event detected in any 5-day period? Is that correct? If that is not the case, I need more clarification as the schematic in Figure A1 is not very detailed. Why is a 65% SIC threshold (L343) chosen for the ice edge mask. Is it arbitrary or based on some other knowledge? How often does the SIC need to drop below 50% to be masked as a polynya (or how many times is "frequently"? L344)?

*The algorithm detects any number of warm air intrusions within the 30-day window. The 5-day period is referring to the time difference between the "new and previous" WAI. Note that the similarity check compares each newly detected WAI with each previously detected WAI. We reformulated the paragraph and hope that it is more clear in the revised version. For example, we will add*
*"The algorithm can detected several not-connected warm air intrusions. If their borders are separated by 2 pixels (= 50 km)*

*or less, the intrusions will be merged.".*

*In L352 to L355 (original manuscript) we describe how the 5-day window method is applied*

100 *"In this study, the algorithm is applied for the winter season November – April. With a 5 day time step, in each iteration, 30 days of T2m and SIC are analyzed"*

*Using 65% as the ice edge detection is somewhat arbitrary. We noticed that analyzing all WAIs, the OSI-SAF SIC never drops below 65% in the affected areas in the central Arctic. Thus this number seemed to be a fair compromise excluding the ice edge where the SIC is generally low and including most of the WAI events. We will add*

105 *"By analyzing all warming events, we found that the sea ice concentration in the Central Arctic from the OSI-SAF algorithm never drops below 65% in the areas affected by an WAI. Thus, this number is a good compromise in order to include as many warm air intrusions as possible while reducing the effect of a moving ice edge on the results." to the manuscript.*

*For the polynia mask, we analyzed 40 years of winter (January - February) SIC and then masked the areas where the SIC was 15% of the times below the threshold of 50%. We only applied this analysis close to the land areas. We will add*

110 *"or the polynya mask, we analyzed 41 years of winter (January and February) OSI-SAF sea ice concentration and masked out areas where the SIC frequently (> 15% of the days) dropped below 50% while the ice edge was far away."*

Comments on Appendix A2: (L364) I assume you are accumulating the area representative of the daily SIC difference from the background SIC over the duration of the WAI event, but you don't specifically state how you compute the effective area loss

115 shown here. Please state specifically how you compute effective area loss. (L369) Is there an average length of WAI events? How was 10 days after the peak warming day chosen? I think revising such that the appendices and the methods are in one section can help with clarifying the above questions.

*L364: Yes, the effective area loss is the sum over the all influenced days (red line in Figure A3). We will add a more detailed*

120 *description to the manuscript*

*"The maximum affected area by this warm air intrusion was around $2 \cdot 10^6\,km^2$. The average sea ice concentration is above 99% before and after the event and drops to 95% during the warm air intrusion (red). The resulting effective area loss, i.e., the sum of the differences between SIC reduced and the average reference SIC, is $6 \cdot 10^5\,km^2$ over the whole red period (16 days) or $\approx 2\%/day$."*

125 *L369: This sentence was not clear enough formulated. The 10 days is referring to the day of maximum SIC reduction and not the temperature peak. We reformulated it to:*

*"If the latter does not apply, the end of the affected period is set to 10 days after the maximum of the SIC reduction. 10 days was chosen by manual analyzing several detected warm air intrusions. We found that most of the times, after 10 days, the effect of warm air intrusions on the sea ice concentration is negligible."*

130

Figure A3: What determines the black portions of the SIC curve where the SIC is below the background value but not included in the WAI event? How is that defined? It's not explained in text. Also, convert the temperature scale to °C to be consistent with the text and other figures.

135 *We will convert the temperature scale to °C and improve the description of the effective area loss calculation given in lines 368 – 373 (original manuscript) to*

*"Defining the time period during which the sea ice concentration is affected by the warm air intrusion is not straight forward. In this study, the following procedure was chosen: The start of the affected period (SIC reduced) is based on the day, when the 2 m air temperature first crossed the $> -10°C$ mark. The end of the period is reached when the sea ice concentration is*

140 *close (within 1%) of the reference sea ice concentration. If the latter does not apply, the end of the affected period is set to 10 days after the maximum of the SIC reduction. 10 days was chosen by manual analyzing several detected warm air intrusions . We found that most of the times, after 10 days, the effect of warm air intrusions on the sea ice concentration is negligible. The reference SIC (blue lines) is the average of the sea ice concentration before and after the warming events. In this example, the drop in SIC around day 10 is related to a different WAI and thus excluded in the analysis."*

145

Technical Corrections

L20: expand the ASI acronym

*Changed*

L33: Warm air intrusions enter the Arctic region, not the Arctic Ocean.

*Changed*

L54: typo – snow/ice

*Corrected*

L58: typo – influences

*Corrected*

L60: The punctuation around the references to papers and figures here is confusing. Please revise.

*We believe that the reviewer refers to " Figure 4 in (Rückert et al., 2023) (see also Figure A7) " and changed it to "Figure B1 (figure 4 from Rückert et al., 2023)"*

L85: typo – daily gridding

*Corrected*

L105-107: This sentence is important, but confusing as written. Please revise to clarify.

*We will rewrite the sentences*
*"An initial temperature threshold is set at* $-10°C$*. In addition, the duration of the wave during which the temperature threshold must be at least two days in order avoid short-term fluctuation around the threshold close in the marginal ice zone."*
*to "... a temperature threshold for an initial detection of a potential warm air intrusion is set at* $-10°C$*. Especially in the marginal ice zone, the 2 m max air temperature can sometimes fluctuate from day-to-day around this threshold even if no WAI is present. In order for the WAI to be further considered, the duration during which this temperature threshold is crossed must be at least two consecutive days."*

L264: extent should be extend

*Corrected*

L289: typo – Another

*Corrected*

L309: revise, "…Arctic amplification have these warm air intrusions increased…"

*We will change the sentence to*
*"II) Did the recent amplified temperature increase in the Arctic led to an increase in frequency and extent of winter warm air*

195    *intrusions? Has the impact of warm air intrusions on satellite sea ice concentration algorithms increased in recent years?"*

Figure 3: Can you make the area notation consistent with the rest of the paper (e.g., 104 km2, etc.)

*Changed*
200

---

## Author Comment (AC2)

**1 Reviewer 2**

Review of "Relevance of warm air intrusions for Arctic satellite sea ice climatologies" by Rostosky and Spreen. https://doi.org/10.5194/tc-2023-69

5    The authors introduce a method to detect and categorize Warm Air Intrusion from atmosphere reanalysis temperature data. They apply the method over four decades and document a (slight) increase in the occurrence and severity of WAIs. They also document that four widely used sea-ice concentration datasets derived from passive microwave data respond differently to WAIs. All datasets are affected by the most severe WAIs (T2m $> -2$C), but three out of four are already significantly affected by less severe WAIs ($-10$C $<$T2m $< -5$C). The study is of interest for users of sea-ice data records, and can be interesting for
10   developers of sea-ice concentration retrieval algorithms. It can be published after some revisions.

*We thank the reviewer for their positive and helpful comments. We believe that, thanks to these reviews, the manuscript has improved a lot. Please find below our responses to the individual comments.*

15   GENERAL COMMENTS:
Kern et al. (2019) concluded that the NSIDC CDR (v3) was high-biased at 100% SIC, while the OSISAF CDR (v2) was low-biased at 100% SIC. This is very well in line with your results. For a SIC CDR to appear un-affected by WAIs, a strategy is to overestimate the SICs (by selecting tiepoints that return $> 100\%$ SICs) and apply the 100% SIC threshold. You see this with your Figure A.6. At present, algorithm calibration is mentioned in passing with only 2 sentences (257-261) in your manuscript,
20   although it is possibly a key element of what you observe. By referring to Kern et al. (2019), using your knowledge of how SIC algorithms work and especially the impact of tie-point selection, and moving your Figure A.6 to the main body of the manuscript, I invite you to significantly expand on these aspects in the Discussions.

*We thank the reviewer for this valuable comment. We added a paragraph do the discussion section and moved figure A6 to*
25   *the main document as suggested by the reviewer. For example, we added*
*"The NSIDC CDR is computed from the NASA-Team and Bootstrap algorithms. The CDR sea ice concentration is based on the sub algorithm with the higher sea ice concentration, which, in case of strong warm air intrusions, is the bootstrap algorithm (since the NASA-Team shows a strong underestimation of sea ice concentration during warm air intrusions). In the NSIDC CDR, an updated bootstrap algorithm with dynamic (daily adapted) tie points for open ocean and full sea ice cover is used*
30   *(Comiso et al., 2017). By using dynamic tie points the impact of changing snow and surface conditions are mitigated and thus the impact of warm air intrusions on the derived sea ice concentration is reduced.*
*While the NSIDC CDR generally performs best during warming events, we note that an overestimation of sea ice concentration can be a result of the method applied in this algorithm. Especially in areas like the Greenland Sea, frequent polynyas and large leads open after strong storm events. These are not captured by the NSIDC CDR, while in the OSI SAF CDR or the ASI*
35   *algorithm using its natural resolution ($6.25 \, \text{km}^2$), these events are clearly visible (see Figure 9). Kern et al., (2019) performed an inter-comparison of several sea ice concentration products and found that the NSIDC CDR systematically overestimates sea ice concentration by around 3% when the ice concentration is close to 100%. This overestimation is not visible in the final product since the NSIDC CDR is truncated at 100% ice concentration. Therefore, in the case of strong warm air intrusions, the NSIDC CDR sea ice concentration can remain close to 100% even though the (non-truncated) average ice concentration*
40   *would drop by a few % (e.g., from 103% to 99%)."*

From your text, it looks like you are using the OSI SAF SIC CDR v2 (OSI-450 and OSI-430-b). If this is the case, I strongly encourage you to update your study to use the SIC CDR v3 (OSI-450-a and OSI-430-a). The v2 CDR is deprecated, the v3 CDR was released in October 2022. Part of the update from v2 to v3 was exactly to reduce the low bias observed in v2 at high
45   SICs. Using the v3 SIC CDR should not be too cumbersome, as it has the same file format and map projection as v2. This would ensure that your manuscript stays relevant for the latest version of the SIC CDRs. If you are already using the v3 CDR, all is good, but you should make it clear in the text (see specific comments below).

*We thank the reviewer for pointing this out. When the study was initialized, OSI-450 was the latest version. In the revised version, we will our analysis with the latest OSI SAF CDR (OSI-450-a). While some numbers changed, the outcome of the study remains the same. In fact, we don't find any improvements using the OSI-450-a product.*

If SIC algorithm developers want to improve their SIC CDRs, they will need to study in details how the Tbs change with WAIs on many WAI cases, across satellite missions, etc... Even if imperfect, your WAI detection method could be a key asset to get these studies started. I thus invite you to: 1) publish the maps of your WAI detection (e.g. daily maps with integer values recording if there was a WAI, and what category it was) and 2) publish the software for the WAI detection on a software repository. This will ensure not only transparency and reproducibility of your research, but also help future developments of SIC algorithms to improve on the situations you report.

*Before the final publication, we plan to publish the code and some working examples on github. Publishing daily maps of detected WAIs would result in a huge dataset. In addition, since the impact of WAIs is lasting for several days to weeks, we believe that such maps would be of limited use. Instead we suggest to add a text file containing monthly statistic of the detected WAIs. In connection with the algorithm, SIC algorithm developers can chose interesting years/events from that text file and apply the detection algorithm for the specific events.*

SPECIFIC COMMENTS:

Title: Your study is only about Sea Ice Concentration (not Drift, Type, etc...). Also, "Climatologies" often refers to the average (or median, etc...) over long time periods (WMO definition). The data you are studying are often referred to as "Climate Data Records", or "timeseries". Please modify the title to reflect the above.

*We will change the title to*
*"Relevance of warm air intrusions for Arctic satellite sea ice concentration time series"*

Abstract :

- "during warm air intrusions". This does not convey that the SIC is impacted after as well (when when Temps are back to nominal).

- The last sentence could be made shorter and more impactful.

*We will change the text to*
*"...during (**and up to 10 days after**) warm air intrusions..."*
*we will change the last sentence to*
*"With a further increase of temperature, such warm air intrusions will occur more frequent and earlier in the season. The influence of these warm air intrusions on sea ice climate data records will therefore become more important in future."*

L23 : You could have cited the recent Kern et al (2019, 2020, 2022) papers. Especially Kern et all 2020 look specifically at summer conditions.

*We thank the reviewer for pointing to these relevant publications and will add them in the introduction section*

L25-26 : Here would be a good place to remind the readers that WAIs have different phases, and that the effects on the microwave emissions might perdure several days after the WAI is finished (in terms of higher temperature).

*That is a good point. We will add*
*"Of importance is that the impact of snow warming, snow metamorphism and snow surface changes due to melt-refreeze events*

*can be visible in the microwave signal for several days and up to weeks after the event (Rückert et al. 2023, e.g.,)"*

L29 : surface ice glazing was the main hypothesis of Rückert et al. (2023) but formulate your sentence as a certainty. You could revise. Also it might be interesting to note that the two case studies in Rtickert et al. (2023) were along the MOSAIC drift.

*We will revise this sentence to:*
*" Rückert et al. (2023) investigated the impact of such a warm air intrusion along the MOSAiC campaign in the central Arctic in April 2020 and found a strong drop in retrieved ice concentration caused by the formation of a large-scale glazed ice layer on top of the snow."*

L33-34 : please specify which versions of the CDRs you are using. Also you should spell out these acronyms.

*We will add the versions used (Version 4 of the NSIDC CRD and version 3 of the OSI SAF CDR) and spelled out the acronyms*

L38 : "the following" — "as follows". Possibly rework these sentences. For example you state here that chapter two will briefly discuss the physics of microwave emissions, but the title of the section is about SIC algorithms, which is not exactly the same thing. Also, I think these are called "Sections" and not "Chapter".

*We will change the sentence to*
*"The article is organized as follows. In section two, the passive microwave sea ice concentration algorithms and auxiliary data used in this study are introduced."*

L45 "quantities" — "surfaces"

*Changed*

L53 You could rework the transition from your first to second paragraph. For example open the 2" with "However, the emissivity of the snow/ice system depends on many parameters". And maybe introduce the WAIs later in the 2" paragraph, when the other parameters have been discussed.

*We will rework this paragraph as suggested by the reviewer*
*"In general, the emissivity of sea ice depends on the physical quantities of the ice and snow as well as on the microwave frequency. In Spreen et al. (2008), Figure 1, the typical emissivity of different surface types (first-year ice, multiyear ice and open ocean) are shown in dependence of typical microwave frequencies used by satellites and most of the common sea ice concentration retrievals. However, the emissivity of the snow/ice system depends on many parameters. The main drivers are snow/ice temperature, ice type and the snow microstructure. Ice layers within or ice crusts at top of the snowpack can influence sea ice concentration retrievals that use polarization differences or ratios (due to their strong impact on horizontal polarization, Comiso et al. (1997); Mätzler et al. (1984)). At frequencies higher than 19 GHz, also parameters like snow grain size and shape become important influences for, e.g, retrievals that use gradient ratios of two different frequencies. Several studies have shown that strong weather events like warm air intrusions, introducing snow metamorphism, melt-refreeze events or liquid water formation in the snow modify the above mentioned parameters and consequently influence the emissivity of the snow/ice system (Liu and Curry, 2003; Rückert et al., 2023; Stroeve et al., 2022; Tonboe et al., 2003, e.g.,). Therefore, warm air intrusions can introduce false changes in the retrieved sea ice concentration (Tonboe et al., 2003, e.g.,)."*

L73, L81, L85, L93, L97, etc... check the numbering of your headings. Should it be "2.1", "2.2", etc...

*We will correct the numbering*

L74 From the URL it seems you are using NSIDC SIC CDR v4. Please check and write it in the text.

*Added*

L81 OSI SAF is prefered to OSI-SAF (throughout the text and figures).

*Changed*

L82 From the URL it seems you are using OSI SAF SIC CDR (OSI-450) and ICDR (OSI-430-b) v2. Check and write it in the text.

*We thank the reviewer from pointing this out. When the study was performed, OSI-450 was the latest version. In the revised manuscript, we will use OSI-450-b*

L82 The OSI SAF processing chain also uses ERAS NWP fields and corrects the Tbs for the atmospheric effects. This could be noted as one could have assumed that this would help in the early phases of the WAI.

*This is an interesting point. We believe that one major outcome of this study is that not only atmospheric, but also surface effects introduced by WAIs can strongly impact sea ice concentration algorithms. We will add*
*"OSI-SAF includes ERA5 reanalysis data for correcting the effect of atmospheric effects on the brightness temperatures."*

L87 I thought ASI was the name of an algorithm. Could you write "ASI SICs from AMSR-E and AMSR2", e.g. ?

*Done*

L97 Since your only auxiliary data is ERA5, you can rename your section.

*Changed*

L98 I think ERA5 (not ERA-5) is the accepted acronyms.

*Changed*

L100 For the ERAS bias, please add some more citations rather than the etc, e.g. Wang et al. 2019, Batrak and Miiller (2019).

*We thank the reviewer for the additional reference and will add them to this section*

L119 the definition of the three categories is not entirely clear. L118 you define T threshold with $> - 10C$, $> -5C$, and $> -2C$. Thus, a category 2 ($> 5C$) is automatically also a category 1 (since $>10C$). But later (L133) you use the thresholds as brackets ( $-10C > T > -5C$). Please clarify in the text and review your figure captions and headings of Table 2.

*We agree that the categories were not defined clearly. We will change the text to*
*"We defined the following categories for the temperature thresholds: category 1: $-10°C < T \leq -5°C$, category 2: $-5°C < T \leq -2°C$, and category 3: $> -2°C$ (in the following, for simplicity we will refer to category 1 as $T > -10°C$ and to category 2 as $T > -5°C$)."*

Fig 1 and Fig 2 : Please use the same y-range for the top row graphs. Currently Fig 1 uses (84-100) while Fig. 2 uses (86-100)

*We now use the same y-range*

Table 2: In Fig 1, Fig 2, and Table 2 you introduce the concept of "All Categories" but this is not explained in the text. Is it the collection of all Cat1, Cat2, and Cat3 events? But it is then not obvious why the number of "All categories" events in Table 2 is not the sum of the three other Categories.

*All categories refers to all the areas where the temperature crossed $-10°C$. It is not necessarily the sum of the individual sub-categories since the effective are reduction is calculated and optimized for all sub-categories and thus the sum of them is expected to be larger than the "all categories" class. We added an explanation to the text*
*"All categories refers to all the areas where the temperature crossed $-10°C$. It is not necessarily the sum of the individual sub-categories since the effective are reduction is calculated and optimized for every individual category."*

Table 2: specify the units of the area and area loss.

*We added the units to the table header*

L180: Rather than performance of the algorithm, you could refer to the impact of the WAIs on the algorithms.

*Changed*

Fig 4 : Fix NSIDC (not NSICD) and OSI SAF (not OSI-SAF).

*Corrected*

Fig3 and Fig 5 : re-use the same colors as Fig 1 and Fig 2 (for the data sources)

*We will now use consistent colors for all figures*

Fig 4: use the same colors as Fig 1 and Fig 2 (Categories)

*We will now use consistent colors for all figures*

Fig 4: add text for the time periods covered in the panels.

*We v now use consistent colors for all figures*

L206: From what I understand, you have two observations: 1) the number / area / category of WAIs detected by your method increases throughout the 40 years, and 2) their impact on the SIC products increases. You might want to reformulate this paragraph to make this distinction.

*We will rework this paragraph to*
*"The results presented in the Figures 6 to 8 show that the strength and frequency of the warm air intrusions increased in the last 20 years, especially during April. Compared to the earlier years between 1980 and 1990, the average area of category 3 warm air intrusions increased from $58 \cdot 10^4$ km to $71 \cdot 10^4$ km in the period from 2010 to 2020. Additionally, the average length of these waves increased from 6 to 8 days (not shown). All of these changes contribute to an increased impact of category 3 warm air intrusions on the sea ice concentration in recent years."*

L218 : The sentence starting with "Through..." seems broken. Please rework.

240 *we will rework this paragraph to*
*"ERA5 2 m air temperature has a known a positive bias over Arctic sea ice (???). Because of this bias, some warm air intrusions might not be captured by the algorithm, even though, in reality, the temperature crossed the defined thresholds. Also, miss-classification could be a result of the temperature bias (e.g., an area which is classified as $> -10°C$ might belong to the $> -5°C$ class in reality)."*

245

L237 : The sentence would work equally well without "sophisticated".

*We will remove "sophisticated"*